# Developing a Revenue Sharing Method for an Operational Transfer-Operate-Transfer Project

**Yanhua Du** [1,2,*] ⬭, **Jun Fang** [1], **Yongjian Ke** [3], **Simon P Philbin** [4] ⬭ **and Jingxiao Zhang** [5,*]

1 School of Civil Engineering and Architecture, Wuhan University of Technology, Wuhan 430070, China; whutfj@126.com
2 School of Civil Engineering and Architecture, Zhengzhou University of Aeronautics, Zhengzhou 450015, China
3 Senior Lecturer, School of Built Environment, University of Technology Sydney, Ultimo, NSW 2007, Australia; yongjian.ke@uts.edu.au
4 Nathu Puri Institute for Engineering and Enterprise, London South Bank University, London SE1 0AA, UK; philbins@lsbu.ac.uk
5 School of Economics and Management, Chang'an University, Xi'an 710064, China
* Correspondence: duyh@zua.edu.cn (Y.D.); zhangjingxiao@chd.edu.cn (J.Z.)

**Abstract:** The transfer-operate-transfer (TOT) project model is used widely as a commercial framework for public-private-partnerships to support provision of infrastructure and enable the delivery of services. However, operational delivery of such projects can encounter certain challenges, such as the need for improved revenue sharing between governmental and private partners. The purpose of this paper is to design a revenue sharing method (RSM) that satisfies the revenue-sharing forecast in the contract design stage and the realized revenue sharing in the contract execution period for an operational TOT project. This approach identifies the impact of external uncertainty and effort level as well as the input ratio on revenue sharing of participants, distributes and reasonably minimizes the project revenue uncertainty among the participants, and achieves an improved matching of the participants' revenue sharing with their risk-sharing, resource input and effort level. The paper utilizes the fuzzy-payoffs Shapley value method for revenue distribution for an operational TOT project, where the fuzzy alliance and input ratio coefficient are adopted to gradually optimize the Shapley value and form the RSM of an operational TOT project. The RSM allows prediction of the revenue sharing of participations under uncertain conditions of project revenue and supports improved decision-making by participants.

**Keywords:** operational TOT project; PPP project; Shapley value; fuzzy payoffs; double-fuzzy revenue sharing

## 1. Introduction

The public-private-partnership (PPP) model has been widely used for providing public infrastructure and services around the world [1]. In this context, a transfer-operate-transfer (TOT) project is one of the common modes of the PPP model, which applies to existing facilities and does not involve the task of construction. In recent years, in order to reduce local governments' debt levels and improve the operational efficiency of existing public services, the Chinese central government has encouraged the transformation of existing public services into cooperation projects between a government and private partner (known as social capital in China). This is achieved through deployment of the TOT model, which involves a private partner participating in the operations of the public service [2]. In this scenario, since the private partner is not a charitable body, it requires a

reasonable return on investment [3–5]. Indeed, the royalty fee of a TOT project acts as a substitute for investment cost [6]. The government determines the franchise period based on the payback period under the minimal internal rate of return (IRR) as expected by the concessionaire [7]. Therefore, the expected revenue of a TOT project during franchise period is the project revenue corresponding to the project company's expected minimal IRR. If the TOT project is fully invested and operated by the private partner, the expected project revenue during the franchise period will completely belong to the private partner. However, if the government participates in the project investment, the expected project revenue during the franchise period needs to be reasonably distributed between the government and the private partner. However, characterized by large-scale investments and long franchise periods, TOT projects generally have high risks accompanied with revenue uncertainty. Since it is affected by the project risk, the IRR and revenue of the TOT project may deviate from expectations; on the other hand, if the effort level of the participants fails to meet expectations, the IRR and revenue of the project will also be lower than expected. Accordingly, the private partner cannot recover its investment, which can thereby trigger renegotiation of the contract and even project failure, such as in the case of the Tianjin Shuanggang Garbage Incineration Power Plant, Jingtong Expressway, and Hangzhou Bay Cross Sea Bridge in China [8]. Furthermore, when the project revenue or IRR is much greater than expected, this will potentially lead to disagreement and conflict among the project stakeholders. For example, the Beijing Capital Airport Expressway, and the Quanzhou Wutong Bridge both failed to provide a reasonable distribution plan due to excessive profits, which resulted in a series of problems such as the government and companies competing for profits, the potential for default, as well as public opposition [9]. Indeed, many PPP projects have encountered disputes or failures since the realized revenues may not be in line with the expectations when the project was initiated. This scenario can lead to an imbalance of revenue distribution between the government and the private partner [10], thereby increasing any renegotiation costs [11], as well as increasing government debt risks for Minimum Revenue Guarantees (MRG) of the PPP project. The payment mechanism of the project is the primary means of enabling an effective revenue-risk distribution [12] and means of incentive for the participants to achieve or exceed the expected revenue and IRR of the project. Therefore, in order to address these problems, there is a need to establish a revenue-sharing method (RSM) that matches revenue uncertainty, effort, and resource input for an operating TOT project jointly invested by government and private partner (hereafter called operating TOT).

In the study of revenue distribution research with revenue uncertainty, there are relatively few researchers investigating the flexible RSM as a way of calculating project total revenue. In the current situation and in order to encourage a private partner to actively participate in PPP projects, government departments provide the MRG or guarantee the minimum return rate to overcome revenue uncertainty in regard to project risk [13–15]. In this scenario, the government department will also need to stipulate the Maximum Revenue Cap (MRC) to prevent the private partner from profiteering [16,17]. The revenue exceeding the MRC level is called excess revenue. Therefore, many scholars have studied the excess revenue distribution of PPP projects. Such scholars believe that the government and private partner should share excess returns according to a specified distribution [11,16,17]. In the case of the revenue distribution model with MRG and MRC, the evaluation standards of MRGs and the MRC become the focus of disputes. Moreover, the MRG increases the risk of the government subsidy expenditure and the level of government debt [11]. This is contrary to the purpose of reducing subsidy expenditure or the debt of the government. Indeed, Abrate et al. [18] used dynamic price variability on revenue maximization and found that higher dynamic price variability leads to higher revenues for hotels. Theoretically, project revenue can be maximized through dynamic price variability of product or service to reduce the risk of government guarantee. However, in general, the price of public facilities influences social interests, and the price adjustment procedure is can be overly complicated, difficult and may not significantly reduce the risk of the government guarantee for the project.

At the same time, revenue uncertainty is one of the main reasons for renegotiation of PPP contracts [19]. PPP projects require far more flexibility and adaptability of contractual relationships

than traditional types of procurement [12,20]. Therefore, in regard to research on avoiding renegotiation for revenue uncertainty, many scholars have sought to achieve the equilibrium of benefits between two parties by adjusting several key variables, such as franchise period, franchise fee, product or service price [21–23]. However, renegotiating or adjusting the franchise period and the franchise fee requires a reassessment of the realized and future project revenue, which can be costly and lacking in flexibility for decision-making [9].

In addition, the uncertainty of effort level not only causes revenue uncertainty, but also affects the revenue-sharing ratio (RSR). However, there is a lack of research on the impact of effort level uncertainty on revenue sharing. Wang et al. [17] pointed out that the effort level of participants is related to the proportion of excess revenue when analyzing the revenue sharing in excess revenue. Some scholars in China also regarded the efforts as an influential factor for the RSR [24–26]. However, insufficient consideration has been given both internationally and in China to the impact of uncertain and/or inadequate effort levels on project revenue and revenue sharing.

Therefore, the purpose of this paper is to design an RSM that satisfies the revenue-sharing forecast in the project design and contract negotiation stage as well as the revenue sharing in the contract execution period of an operational TOT project. The method deployed emphasizes the impact of certain effort levels on revenue sharing, increases the flexibility of the contract, avoids the governments' debt risk increasing due to the MRG, reduces the likelihood of renegotiations, and minimizes the project revenue uncertainty among the participants reasonably, which is caused by the external uncertainty and effort level uncertainty. Consequently, an improved matching can be achieved of the operational TOT project revenue sharing with risk sharing, resource input and effort level. This should have the effect of encouraging the participants improve their effort levels for the project.

The main contribution of this research study has three aspects. (1) The paper proposes a concise algorithm for the input ratio correction coefficient, which simplifies the method and supports further method adoption. (2) The double-fuzzy Shapley value method proposed in this paper predicts the revenue sharing of participants under project revenue uncertainty. This enables participants to be clear on the impact of project revenue uncertainty on revenue sharing and make more reasonable decisions in the contract design stage. (3) The revenue uncertainty is reasonably apportioned among the participants in proportion through this method to avoid the possible dispute of determining the MRG and MRC.

## 2. Literature Review

### 2.1. Common Methods of Revenue Sharing in PPP Projects

Kang et al. [27] believe that the PPP contract itself can be regarded as a scheme to enable revenue sharing between the government and the private sector. Indeed, many PPP contracts contain revenue-sharing clauses [28,29], and some general contracts also use revenue-sharing contracts [30]. Game theory is a major way to study revenue-sharing contracts [31]. Kang et al. [27,32] developed a heuristic algorithm for the bi-level programming problem involving the government and the private partner, and they reached a revenue-sharing scheme between the government and the private partner through the process of bargaining. However, the learning effect of this method must be determined subjectively, which influenced its objectivity to some extent. Wang et al. [17] developed an approach that involved excess revenue sharing between the government and investors, which was based on incentive and principal-agent theories. The authors also believed that the sharing ratio for excess revenue was related to the fairness preferences and the effort cost coefficient of the investors. Alas, they only allocated excess revenue without considering the total project revenue. Furthermore, Shang et al. [33] established a bargaining model for energy-saving revenue distribution using Rubinstein's bargaining game theory. The researchers obtained an effective bargaining interval that was satisfactory to both parties through the game, but the solution was not unique. The non-uniqueness and uncertainty of the Nash bargaining model solution can be considered as a limitation of this method.

There are other revenue distribution methods. For instance, Carbonara et al. [34] proposed a model for assessing and benchmarking the net benefits of the different Energy Performance Contracting (EPC) structures. They used the net present value (NPV) differential minimization method to balance the profit demand of the private sector and the economic interests of the public sector for selecting the EPC schema, which thereby creates a "win-win" solution for both parties. However, it can be observed that there were insufficient considerations about how much the participants contribute to the process.

## 2.2. Shapley Value Evolution and Its Application in TOT Project Revenue Sharing

Shapley value is a method proposed by L.S. Shapley in 1953 in order to solve the cost-sharing revenue distribution of alliance members in classical cooperative games [35]. This approach provides a strict axiomatic description to subjective concepts, such as "fairness" or "reasonableness", and is an established way to study the fair revenue distribution within the alliance [36]. Later, many scholars studied the Shapley value for the scenario of the fuzzy cooperative game. On the one hand, Shapley value is studied under fuzzy alliance. In this regard, Aubin [37] formally proposed the concept of fuzzy alliance and fuzzy game in 1974, that is, the players participate in multiple alliances with a participation rate between 0–1 as a fuzzy game with a fuzzy alliance. In 2001, Tsurumi [38] proposed to calculate Shapley under fuzzy union conditions using the Choquet integral. Furthermore, Shapley value can be studied under the fuzzy condition of payoff function. Mares [39] proposed a cooperative strategy with fuzzy payoffs. Chen et al. [40] extended the axiom of the classical Shapley value to Shapley value with interval fuzzy payoff by fuzzy set theory. Zhao et al. [41] extended the participation of fuzzy alliance to fuzzy value and proposed the triangular fuzzy structure element expression of the double-fuzzy Shapley value of fuzzy alliance and fuzzy payoffs combination. The above theoretical research of Shapley value provides theoretical support for the analysis of the impact of revenue uncertainty on revenue sharing for the TOT project.

As for the application research of the Shapley value method for the case of PPP project revenue sharing, many scholars have proposed multi-factor correction of the Shapley value, as detailed in Table 1. Hu et al. [42] established a modified Shapely value PPP project revenue distribution model, taking into account investment proportion, risk allocation coefficient, contract execution degree and contribution degree. Li et al. [43] proposed a Shapley value correction model for sewage treatment projects that took into account risk factors, contribution of investment and contribution of innovation ability to ensure the fairness of income distribution. The method of multi-factor correction of Shapley value usually uses the product of the TOT project revenue and the difference between one factor weight and 1/n to adjust the revenue sharing one by one. The calculation method is relatively cumbersome. In addition, the method of multi-factor correction of Shapley value fails to consider the impact of project revenue uncertainty on revenue sharing. Yu et al. [44] introduced uncertainty into the revenue distribution model and proposed the interval Shapley value method to obtain the participants' fuzzy revenue-sharing interval. However, the projects' revenue uncertainty and the participants' revenue-sharing were not sufficiently matched. Also, Zhang [45] respectively applied fuzzy-payoffs Shapley value and fuzzy-alliance Shapley value to study the revenue distribution of contract energy management projects but were not able to combine fuzzy payoffs and fuzzy alliance to solve the revenue distribution.

## 2.3. Main Influencing Factors of Revenue Sharing for TOT Projects

The revenue sharing is the product of the project revenue and the RSR, where the project revenue can be affected by external uncertainty and the effort level of participants [46]. The RSR is mainly related to the effort level and input ratio of the participants in the project [17,24]. The effort level not only affects the project revenue but also affects the RSR. The uncertainty about the effort level will lead to changes in project revenue and RSR, which is different from the external uncertainty and the input ratio. Therefore, this paper divides the factors affecting the revenue sharing into three indicators, namely: external uncertainty of the project, effort level, and input ratio. External uncertainties are mainly caused by total project risks. The effort level is mainly manifested in three aspects: contract

execution degree, undertaking task complexity and mutual satisfaction. The resource input ratio consists of investment proportion, risk-sharing proportion, innovation investment proportion and critical problem investment proportion.

**Table 1.** Modifying factors and disadvantage of different modified Shapley Value methods.

| Research Study | Modifying Factors | Unconsidered Factors |
|---|---|---|
| Hu et al. [42] | Investment proportion, risk allocation, contract execution degree, contribution degree. | Contribution of innovation revenue uncertainty, the uncertainty of effort level. |
| Li et al. [43] | Risk allocation, investment proportion, contribution of innovation. | Contract execution degree revenue uncertainty, the uncertainty of effort level. |
| Yu et al. [44] | Revenue uncertainty, investment proportion, risk allocation, contract execution degree. | Contribution of innovation, contract execution degree the uncertainty of effort level. |
| Zhang [45] | Revenue uncertainty, or participation rate less than 1. | Investment proportion, risk allocation, contract execution degree. |

In summary, very few scholars have explored how to distribute the operational TOT project revenue by revising the Shapley value method based on project external uncertainty, effort level and resource input ratio. Therefore, this paper analyzes revenue sharing on TOT projects by different methods and from the perspectives of external uncertainty, effort level and input ratio. Additionally, the paper constructs an RSM of the operational TOT project based on input ratio and double-fuzzy Shapley value in order to realize revenue sharing matching with project risk-sharing, effort level and resource input ratio of the participants.

## 3. Methods

### 3.1. Research Design

This research study assumes that the project revenue changes along with external uncertainties and the effort level of participants, and that the total resource input of the project is unchanged.

Due to the external uncertainty of the operational TOT project, the different input ratio of participants, and the failure to reach the expected level of effort, this study selects the modified Shapley value method to construct the RSM of the operational TOT project according to the findings from the literature review. The classic Shapley value is modified in three steps in combination with the three main influencing factors analyzed above. Firstly, an RSM based on fuzzy payoff Shapley value (hereafter called Method #1) is constructed to analyze the influence of revenue uncertainty on revenue sharing. Secondly, on the basis of Method #1, Shapley value is modified again by fuzzy alliance for effort level less than expected and its uncertainty, and then an RSM based on double-fuzzy Shapley value (hereafter called Method #2) is proposed for the impact of effort level less than expected and its corresponding uncertainty on project revenue and RSR. Finally, the correction coefficient of input ratio is proposed to further optimize the Method #2 for the impact of the participants' input ratio on revenue sharing, and finally it constructs an RSM of an operational TOT project based on the input ratio and double-fuzzy Shapley value (hereafter called Method #3).

The range of revenue uncertainty caused by external uncertainty is determined according to the risk assessment data for specific projects. The scores of the secondary indicators of effort level and input ratio are compiled through specific project data, and the weights of the secondary indicators are obtained by the expert scoring method as shown in Section 3.2 related calculations.

### 3.2. Parameter Calculation of the Effort Level and Input Ratio

#### 3.2.1. Calculation of the Effort Level

The secondary indicators of the effort level for an operational TOT project are contract execution degree, undertaking task complexity and mutual satisfaction, which are respectively denoted as $R_{11}$, $R_{12}$, $R_{13}$. The scores of each indicator are $U_{i1}$, $U_{i2}$, $U_{i3}$, and the weights of each indicator are respectively denoted as $w_{11}$, $w_{12}$ and $w_{13}$. Then, the expression of effort level is:

$$U_i = \sum w_{11}U_{i1} + w_{12}U_{i2} + w_{13}U_{i3} \tag{1}$$

The weight of the secondary indicators was determined by the expert scoring method, the score of the mutual satisfaction index is obtained through the questionnaire, and the scores of other indexes are obtained through the relevant data from the operational management of the project. In this research study, 16 experts were invited to use the coercive 0–4 scoring method to score the weight of the secondary indicators of effort level respectively. The highest score and the lowest score are removed from each index score, and the total score is shown in Table 2. The weight calculation process of each index is shown in Table 3. According to Table 3, $w_{11}$ = 0.45, $w_{12}$ = 0.24, and $w_{13}$ = 0.31.

**Table 2.** Expert scoring and score summary of effort level secondary indicators.

| Indicators | 0 | 1 | 2 | 3 | 4 | Number of Experts | Score |
|---|---|---|---|---|---|---|---|
| Importance of $R_{11}$ relative to $R_{12}$ | 0 | 1 | 4 | 9 | 2 | 16 | 39 |
| Importance of $R_{11}$ relative to $R_{13}$ | 0 | 2 | 4 | 8 | 2 | 16 | 37 |
| Importance of $R_{12}$ relative to $R_{13}$ | 1 | 6 | 7 | 2 | 0 | 16 | 23 |

**Table 3.** Weight calculation of secondary index of effort level.

| Indicators | $R_{11}$ | $R_{12}$ | $R_{13}$ | Score | Corrected Score | Weights |
|---|---|---|---|---|---|---|
| $R_{11}$ | - | 39 | 37 | 77 | 78 | 0.45 |
| $R_{12}$ | 17 | - | 23 | 40 | 41 | 0.24 |
| $R_{13}$ | 19 | 33 | - | 42 | 43 | 0.31 |
| Total | | | | 169 | 172 | 1 |

#### 3.2.2. Calculation of the Input Ratio

The input ratio includes four indicators, namely, the investment proportion, risk-sharing proportion, innovation investment proportion and critical problem investment proportion, which are respectively written as $R_{21}$, $R_{22}$, $R_{23}$ and $R_{24}$, the scores of the secondary index are written as $\eta_{i1}$, $\eta_{i2}$, $\eta_{i3}$, $\eta_{i4}$, and the weights of the secondary index are written as $w_{21}$, $w_{22}$, $w_{23}$ and $w_{24}$. Then, the correction coefficient of input ratio is expressed as:

$$\eta_i = \sum w_{21}\eta_{i1} + w_{22}\eta_{i2} + w_{23}\eta_{i3} + w_{24}\eta_{i4} \tag{2}$$

The secondary index scores can be obtained through the contract and relevant data of the operational management of the project. The weights of the secondary index are determined by the expert scoring method. The specific process is similar to the expert scoring method in Section 3.2.1. The scoring and calculation procedures are shown in Tables 4 and 5.

**Table 4.** Expert scoring and score summary of input ratio secondary indicators.

| Indicators | 0 | 1 | 2 | 3 | 4 | Number of Experts | Score |
|---|---|---|---|---|---|---|---|
| Importance of $R_{21}$ relative to $R_{22}$ | 0 | 2 | 7 | 4 | 3 | 16 | 35 |
| Importance of $R_{21}$ relative to $R_{23}$ | 1 | 4 | 6 | 3 | 2 | 16 | 29 |
| Importance of $R_{21}$ relative to $R_{24}$ | 0 | 2 | 7 | 5 | 2 | 16 | 34 |
| Importance of $R_{22}$ relative to $R_{23}$ | 0 | 2 | 8 | 4 | 2 | 16 | 33 |
| Importance of $R_{22}$ relative to $R_{24}$ | 0 | 2 | 7 | 5 | 2 | 16 | 34 |
| Importance of $R_{23}$ relative to $R_{24}$ | 0 | 4 | 6 | 4 | 2 | 16 | 31 |

**Table 5.** Weight calculation of secondary index of input ratio.

| Indicators | $R_{21}$ | $R_{22}$ | $R_{23}$ | $R_{24}$ | Score | Corrected Score | Weights |
|---|---|---|---|---|---|---|---|
| $R_{21}$ | - | 35 | 29 | 34 | 98 | 99 | 0.29 |
| $R_{22}$ | 21 | - | 33 | 34 | 88 | 89 | 0.26 |
| $R_{23}$ | 27 | 23 | - | 31 | 81 | 82 | 0.24 |
| $R_{24}$ | 22 | 22 | 25 | - | 69 | 70 | 0.21 |
| Total | | | | | 336 | 340 | |

According to Table 5, $w_{21} = 0.29$, $w_{22} = 0.26$, $w_{23} = 0.24$, and $w_{24} = 0.21$.

### 3.3. Development of the Operational TOT Project RSM

### 3.3.1. Relevant Concepts

**Definition 1.** *Fuzzy payoff Shapley value of the operational TOT project.*

According to the Shapley value theory, $(N, \widetilde{v})$ is the fuzzy cooperative game of operational TOT project, where $\widetilde{v} : P(N) \rightarrow \widetilde{R}_+$ and $\widetilde{v}(\phi) = 0$, and the fuzzy payoffs Shapley value is as follows [41]:

$$\widetilde{\varphi}_i(\widetilde{v}) = \sum_{i \in S} \frac{(|S|-1)!(2-|S|)!}{2!}(\widetilde{v}(S) - \widetilde{v}(S\backslash i)) \tag{3}$$

This Shapley value does not represent revenue sharing, but only represents the revenue sharing proportion. The latter value of Shapley is the same as above.

**Definition 2.** *Double-fuzzy payoff function and double-fuzzy Shapley value of the operational TOT project.*

According to the definition of fuzzy alliance [37], the fuzzy alliance of an operational TOT project means that the participants in the project participate in the alliance with an effort level between 0–1. In this research study, the payoff function of the fuzzy cooperative game with fuzzy alliance and fuzzy payoff (short for double-fuzzy cooperative game) in an operational TOT project is as follows [37]:

$$\widetilde{v}_{FF}(\widetilde{S}) = \int \widetilde{S} d\widetilde{v} = \sum_{l=1}^{m(\widetilde{S})} \widetilde{v}\left(\left[\widetilde{S}\right]_{\overline{h}_l}\right)\left(\widetilde{h}_l - \widetilde{h}_{l-1}\right) \tag{4}$$

The double-fuzzy Shapley value of the operational TOT project is as follows [41]:

$$\widetilde{\widetilde{\varphi}}_i(\widetilde{v})(U) = \sum_{l=1}^{m(U)} \widetilde{\varphi}_i(\widetilde{v})\left([U]_{\overline{h}_l}\right)\left(\widetilde{h}_l - \widetilde{h}_{l-1}\right) \tag{5}$$

**Definition 3.** *Double-fuzzy Shapley value of the operational TOT project based on input ratio.*

The cooperation strategy based on investment proportion correction is denoted as $\widetilde{v}^{\eta} : \widetilde{G}_{FF}(N) \to R$, and $\widetilde{v}^{\eta}(\phi) = 0$, then the double-fuzzy Shapley value of the operational TOT project based on input ratio is as follows:

$$\widetilde{\varphi}_i^{\eta}(\widetilde{v})(U) = 2\eta_i \widetilde{\varphi}_i(\widetilde{v})(U) \quad (i = g, c) \tag{6}$$

**Definition 4.** *Structural element representation of fuzzy payoff and fuzzy effort levels.*

Let $E$ be a triangular symmetric fuzzy structural element [40,41]. The value interval is $E = [-1, 1]$, and its membership function is as follows:

$$E(x) = \begin{cases} 1 + x, -1 \le x \le 0 \\ 1 - x, 0 < x \le 1 \\ 0, \quad else \end{cases}$$

Let $f$ and $k$ be the monotone function with the same formal on interval $E = [-1, 1]$. $\widetilde{v}(S) = f_S(E)$, $\widetilde{v}_{FF}(\widetilde{S}) = f_{\widetilde{S}}(E)$, $\widetilde{h}_l = k_l(E)$.

Note $f(x) = ax + b$, $k(x) = cx + d$.

The fuzzy number generated by the linear fuzzy structure element is: $\widetilde{v} = a + bE$, $\widetilde{h} = c + dE$. Then, $\widetilde{v} = f(E)$, $\widetilde{h} = k(E)$.

### 3.3.2. RSM of the Operational TOT Project Based on Fuzzy Payoff Shapley Value

According to Definition 4, the structure element linear expression of expected revenue of the operational TOT project based on fuzzy payoff is:

$$\widetilde{v}_{TOT} = a_{TOT} + b_{TOT}E \tag{7}$$

According to Formula (3) and Definition 4, the fuzzy payoff Shapley value of the operational TOT project is:

$$\widetilde{\varphi}_i(E) = \sum_{\widetilde{S} \subseteq N} \frac{(|S| - 1)!(2 - |S|)!}{2!} \left[ a_S - a_{S\setminus i} + (b_S + b_{S\setminus i})E \right] \tag{8}$$

The RSR of the participants, which is based on fuzzy payoff are:

$$\widetilde{\beta}_i(E) = \frac{\widetilde{\varphi}_i(E)}{\sum \widetilde{\varphi}_i(E)} \tag{9}$$

The revenue sharing of the participants based on fuzzy payoff are:

$$\widetilde{\gamma}_i = \widetilde{\beta}_i \widetilde{v}_{TOT} \tag{10}$$

### 3.3.3. RSM of the Operational TOT Project Based on Double-Fuzzy Shapley Value

According to Formula (4) and Definition 4, the structural element linear expression of the double-fuzzy payoffs function is:

$$\widetilde{v}_{FF}(\widetilde{S}) = \sum_{l=1}^{m(\widetilde{S})} (a_{\widetilde{S}} + b_{\widetilde{S}}E)[c_l - c_{l-1} + (d_l + d_{l-1})E] \tag{11}$$

According to Formula (5) and Definition 4, the structure element linear expression of the double-fuzzy Shapley value of the operational TOT project is:

$$\widetilde{\widetilde{\varphi}}_i(E) = \sum_{l=1}^{m(\widetilde{S})} \left[ \left( \sum_{\widetilde{S} \subseteq N} \widetilde{\varphi}_i(E) \right) [H]_{\widetilde{h}_l} \right] \times [c_l - c_{l-1} + (d_l + d_{l-1})E] \tag{12}$$

The RSR of the participants, which is based on double-fuzzy Shapley value are:

$$\widetilde{\widetilde{\beta}}_i(E) = \frac{\widetilde{\widetilde{\varphi}}_i(E)}{\sum \widetilde{\widetilde{\varphi}}_i(E)} \tag{13}$$

The revenue sharing of the participants based on double-fuzzy Shapley value are:

$$\widetilde{\widetilde{\gamma}}_i = \widetilde{\widetilde{\beta}}_i \widetilde{\widetilde{v}}_{TOT} \tag{14}$$

where, $\widetilde{\widetilde{v}}_{TOT}$ represents the expected revenue of the operational TOT project based on double-fuzzy Shapley value.

### 3.3.4. RSM of the Operational TOT Project Based on Input Ratio and Double-Fuzzy Shapley Value

Project revenue does not change with the input ratio of participants, so $\widetilde{\widetilde{v}}_{TOT}^{\eta} = \widetilde{\widetilde{v}}_{TOT}$. Where, $\widetilde{\widetilde{v}}_{TOT}^{\eta}$ is the expected revenue of the operational TOT project based on input ratio and double-fuzzy Shapley value.

According to Formula (6) and Definition 4, the structure element linear expression of the double-fuzzy Shapley value of operational TOT project based on the input ratio is:

$$\widetilde{\widetilde{\varphi}}_i^{\eta}(E) = 2\eta_i \widetilde{\widetilde{\varphi}}_i(\widetilde{v})(H) = 2\eta_i \sum_{l=1}^{m(\widetilde{S})} \left[ \left( \sum_{\widetilde{S} \subseteq N} \widetilde{\varphi}_i(E) \right) [U]_{\widetilde{h}_l} \right] \times [c_l - c_{l-1} + (d_l + d_{l-1})E] \tag{15}$$

The RSR of the participants, which is based on input ratio and double-fuzzy Shapley value are:

$$\widetilde{\widetilde{\beta}}_i^{\eta}(E) = \frac{\widetilde{\widetilde{\varphi}}_i^{\eta}(E)}{\sum \widetilde{\widetilde{\varphi}}_i^{\eta}(E)} \tag{16}$$

The revenue sharing of the participants, which is based on input ratio and double-fuzzy Shapley value are:

$$\widetilde{\widetilde{\gamma}}_i^{\eta} = \widetilde{\widetilde{\beta}}_i^{\eta}(E) \times \widetilde{\widetilde{v}}_{TOT}^{\eta} \tag{17}$$

## 4. Case Study

### 4.1. Background of the Case

The Laohekou City Funeral Service Center in Hubei Province of China is an existing operational infrastructure that is planning to adopt the TOT project model. It is a typical operational TOT project with an investment return mechanism using the "user pays" approach. The project location is shown in Figure 1 and as depicted at the yellow star.

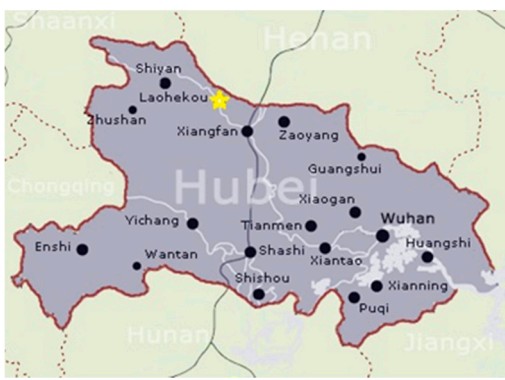

**Figure 1.** The location of the operational transfer-operate-transfer (TOT) project.

In the operational TOT project, a Cemetery Management Office in Laohekou City acts as a government authorized investment representative (hereinafter referred to as the government) which jointly establishes a project company with private partner. The project company will purchase the existing project assets with RMB 115.9 million and invest RMB 3 million as the project liquidity. The government contributes approximately RMB 7.1 million, accounting for 20% of the capital; the private partner contributes approximately RMB 28.6 million, accounting for 80% of the capital. The cooperative operational TOT project has a franchise period of 15 years.

According to the data of the "implementation plan of the PPP project of the Laohekou City funeral service" and the "value-for-money evaluation plan of the Laohekou funeral service PPP project", regardless of the time value of funds, the total net profit of the project during the franchise period is RMB 46.06 million in the TOT mode. The total net profit of the project during the franchise period is RMB 28.34 million in the traditional mode. If the project is implemented by the private partner, the total net profit of the project during the franchise period is RMB 45.59 million according to the reasonable calculation of the relevant data of the traditional mode and TOT project mode.

In the research study it is assumed that the range of revenue uncertainty caused by external uncertainty is ±10%; and the effort levels fluctuate ±0.05. The scores of each index of the government and private partner are shown in Table 6. The scores in blue of Table 6 represent the assumed value.

**Table 6.** The scores of each index of the government and private partner.

|  | Indicators | Government | Private Partner |
|---|---|---|---|
| Effort Level | Contract execution degree | 1 | 0.75 |
| | Undertaking task complexity | 0.7 | 1 |
| | Mutual satisfaction | 0.9 | 0.7 |
| Input Ratio | Investment proportion | 0.2 | 0.8 |
| | Risk-sharing proportion | 0.3 | 0.7 |
| | Innovation investment proportion | 0.1 | 0.9 |
| | Critical problem investment proportion | 0.25 | 0.75 |

*4.2. Revenue Sharing of the Operational TOT Project Participants*

4.2.1. Revenue Sharing of the Operational TOT Project with Method #1

According to the case data and definition 4, $f_g(E) = 28.34 + 2.83E$, $f_c(E) = 44.59 + 4.46E$, $f_{TOT}(E) = 46.06 + 4.61E$, $\widetilde{v}_{TOT} = 46.06 + 4.61E$. The interval of $\widetilde{v}_{TOT}$ is RMB 41.45–50.66 million.

According to Formula (8), $\widetilde{\varphi}_g\{g,c\} = 14.90 + 5.95E$, $\widetilde{\varphi}_c\{g,c\} = 31.15 + 5.95E$.

According to Formula (9), $\widetilde{\beta}_g = \frac{14.90+5.95E}{46.06+11.90E}$, $\widetilde{\beta}_c = \frac{31.15+5.95E}{46.06+11.90E}$. The interval of $\widetilde{\beta}_g$ is [26.21%, 35.98%], and the interval of $\widetilde{\beta}_c$ is [73.79%, 64.02%].

According to Formula (10), $\widetilde{\gamma}_g = \frac{14.90+5.95E}{46.06+11.90E}(46.06 + 4.61E)$, $\widetilde{\gamma}_c = \frac{31.15+5.95E}{46.06+11.90E}(46.06 + 4.61E)$. The interval of $\widetilde{\gamma}_g$ is RMB 10.87–18.23 million, and the interval of $\widetilde{\gamma}_c$ is RMB 30.59–32.43 million.

### 4.2.2. Revenue Sharing of the Operational TOT Project with Method #2

According to Formula (1) and case data, $U_g = 0.9$, and $U_c = 0.8$, $k_g(E) = 0.9 + 0.05E$, and $k_c(E) = 0.8 + 0.05E$. Therefore, $\widetilde{h}_1 = k_c(E)$ and $\widetilde{h}_2 = k_g(E)$.

According to Formula (11), $\widetilde{\widetilde{v}}_{TOT} = 0.51E^2 + 9.11E + 39.68$. The interval of $\widetilde{\widetilde{v}}_{TOT}$ is RMB 31.09–49.30 million.

According to Formula (12), $\widetilde{\widetilde{\varphi}}_c = 29.75E^2 + 631.75E + 2492.38$, $\widetilde{\widetilde{\varphi}}_g = 58.09E^2 + 826.26E + 1475.73$.

According to Formula (13), $\widetilde{\widetilde{\beta}}_g = \frac{58.09E^2+862.26E+1475.73}{87.84E^2+1494.01E+3968.10}$, $\widetilde{\widetilde{\beta}}_c = \frac{29.75E^2+631.75E+2492.38}{87.84E^2+1494.01E+3968.10}$.

The interval of $\widetilde{\widetilde{\beta}}_g$ is 26.21–43.17%, and the interval of $\widetilde{\widetilde{\beta}}_c$ is 73.79–56.83%.

According to Formula (14), $\widetilde{\widetilde{\gamma}}_g = \frac{0.58E^2+8.62E+14.76}{0.88E^2+14.94E+39.68}(0.51E^2 + 9.11E + 39.68)$, $\widetilde{\widetilde{\gamma}}_c = \frac{0.30E^2+6.32E+24.92}{0.89E^2+14.94E+39.68}(0.51E^2 + 9.11E + 39.68)$. The interval of $\widetilde{\widetilde{\gamma}}_g$ is RMB 8.15–21.28 million, and the interval of $\widetilde{\widetilde{\gamma}}_c$ is RMB 22.94–28.02 million.

### 4.2.3. Revenue Sharing of the Operational TOT Project with Method #3

$\widetilde{\widetilde{v}}_{TOT}^{\eta} = \widetilde{\widetilde{v}}_{TOT} = 0.51E^2 + 9.11E + 39.68$. The interval of $\widetilde{\widetilde{v}}_{TOT}^{\eta}$ is RMB 31.09–49.30 million.

According to Formula (2) and the date in Table 1, $\eta_g = 0.21$, and $\eta_c = 0.79$.

According to Formula (15), $\widetilde{\widetilde{\varphi}}_g^{\eta} = 6.20 + 3.62E + 0.24E^2$, and $\widetilde{\widetilde{\varphi}}_c^{\eta} = 39.38 + 9.88E + 0.47E^2$.

According to Formula (16), $\widetilde{\widetilde{\beta}}_g^{\eta}(E) = \frac{6.20+3.62E+0.24E^2}{45.58+13.60E+0.71E^2}$, and $\widetilde{\widetilde{\beta}}_c^{\eta}(E) = \frac{39.38+9.88E+0.47E^2}{45.58+13.60E+0.71E^2}$. The interval of $\widetilde{\widetilde{\beta}}_g^{\eta}$ is 8.63–16.80%, and the interval of $\widetilde{\widetilde{\beta}}_c^{\eta}$ is 91.37–83.20%.

According to Formula (17), $\widetilde{\widetilde{\gamma}}_g^{\eta} = \frac{6.20+3.62E+0.24E^2}{45.58+13.60E+0.71E^2}(39.68 + 9.11E + 0.51E^2)$, and $\widetilde{\widetilde{\gamma}}_c^{\eta} = \frac{39.38+9.88E+0.47E^2}{45.58+13.60E+0.71E^2}(39.68 + 9.11E + 0.51E^2)$. The interval of $\widetilde{\widetilde{\gamma}}_g^{\eta}$ is RMB 2.68–8.28 million, and the interval of $\widetilde{\widetilde{\gamma}}_c^{\eta}$ is RMB 28.41–41.02 million.

## 5. Results and Analysis

The results and analysis involve comparison of the results of three RSMs, and the potential application function of Method #3, as well as the comparison of different modified Shapley value methods. Comparison 1 is the comparison of Method #1 with the classical Shapley value method to analyze the impact of external uncertainty on revenue sharing. Comparison 2 is a comparison of Method #2 with Method #1 in order to analyze the impact of effort level and its uncertainty on the project revenue and the RSR. Comparison 3 is the comparison of Method #3 with Method #2 to analyze the impact of input ratio on revenue sharing.

### 5.1. Comparison 1

When $E = 0$ in Method 1, the results are the revenue-sharing values based on the classic Shapley value. The revenue-sharing of Method #1 is the interval value of $E = [-1, 1]$, including $E = 0$. Therefore, the revenue sharing based on the classic Shapley value is a special point of the Method #1. The revenue interval value of the project indicates the uncertainty range of the project revenue caused by external uncertainty. The revenue-sharing interval value of the participant is the revenue-sharing range corresponding to the project revenue uncertainty. Consequently, the uncertainty of project revenue is reasonably distributed between the participants according to the respective RSRs.

*5.2. Comparison 2*

In Method #1, the efforts of the participants are equal to 1; while in Method #2, $\widetilde{h}_c = [0.75, 0.85]$, and $\widetilde{h}_g = [0.85, 0.95]$.

5.2.1. Changes of Government Revenue Sharing Caused by Effort Level

Figure 2 shows that the project revenue decreases due to the decreased effort level of the participants. While the RSR of the government increases for $\widetilde{h}_g > \widetilde{h}_c$ as shown in Figure 3, and when the $E$ value is low, the revenue sharing decreases, when the $E$ value is high, the revenue sharing increases as shown in Figure 4. Comparing with Method #1, Method #2 shows that the project revenue decreases due to the decrease of effort level, and the RSR increases for $\widetilde{h}_g > \widetilde{h}_c$, however, the revenue sharing is uncertain.

The data in Table 7 shows that, when $E = 0.2$, the effort level of the government decreases by 9%, the project revenue decreases by RMB 5.46 million, a decrease of 11.62%; while $E = 0.2$, the effort level of the government decreases by 11%, and the project revenue decreases by RMB 7.26 million with a decrease of 16.08%. Therefore, the expected revenue of the operational TOT project is positively correlated with the effort levels of the government.

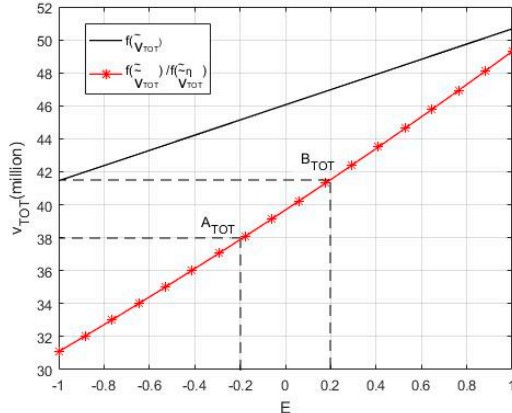

**Figure 2.** Comparison of $\widetilde{v}_{TOT}$ and $\widetilde{\widetilde{v}}_{TOT}$ ($\widetilde{\widetilde{v}}_{TOT}^{\eta}$).

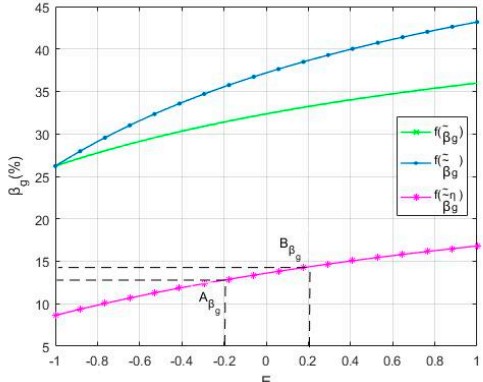

**Figure 3.** Comparison of $\widetilde{\beta}_g$, $\widetilde{\widetilde{\beta}}_g$ and $\widetilde{\widetilde{\beta}}_g^{\eta}$.

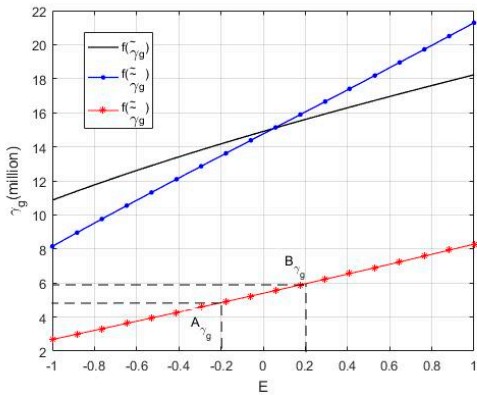

**Figure 4.** Comparison of $\widetilde{\gamma}_g$, $\widetilde{\widetilde{\gamma}}_g$ and $\widetilde{\widetilde{\gamma}}_g^{\eta}$.

**Table 7.** Comparison of the revenue sharing of the government between Method #1 and Method #2 ($E = \pm 0.2$).

| Indicators | Effort Level | | Project Revenue | | Revenue-Sharing Ratio (RSR) | | Revenue Sharing | |
|---|---|---|---|---|---|---|---|---|
| | $E = 0.2$ | $E = -0.2$ | $E = 0.2$ | $E = -0.2$ | $E = 0.2$ | $E = -0.2$ | $E = 0.2$ | $E = -0.2$ |
| Method #1 | 1 | 1 | 46.98 | 45.14 | 33.23% | 31.40% | 15.61 | 14.17 |
| Method #2 | 0.91 | 0.89 | 41.528 | 37.88 | 38.65% | 35.55% | 16.05 | 13.47 |
| Comparison | −0.09 | −0.11 | −5.46 | −7.26 | 5.42% | 4.15% | 0.44 | −0.70 |
| Comparison (%) | −9% | −11% | −11.62% | −16.08% | - | - | 5.80% | −4.98% |

When $E = 0.2$, the RSR for the government increased by 5.42%, and the revenue sharing increased by RMB 0.44 million, which is an increase of 2.80%. When $E = -0.2$, its RSR increased by 4.15%, and the revenue sharing decreased by RMB 0.71 million, which is a decrease of 4.98%. This indicates that RSR is positively correlated with $E$, and the project revenue is higher, and the revenue sharing is increased when $E$ is larger, the project revenue is lower, and the revenue sharing is reduced when $E$ is smaller. Therefore, in order to maximize the benefits, participants not only should increase their RSRs, but also cooperate with each other in order to increase project revenue, which emphasizes the importance of cooperation in an operational TOT project.

5.2.2. Changes of Private Partner Revenue Sharing Caused by Effort Level

Figures 2, 5 and 6 show that the project revenue decreases, and the RSR of the private partner decreases for $\widetilde{h}_c < \widetilde{h}_g$, Moreover, the comparison of Method #2 with Method #1 highlights that the project revenue and the RSR change in the same direction, and that the revenue share changes more sharply in the same direction.

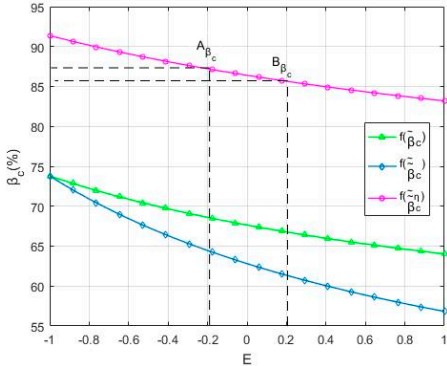

**Figure 5.** Comparison of $\widetilde{\beta}_c$, $\widetilde{\widetilde{\beta}}_c$ and $\widetilde{\widetilde{\beta}}_c^{\eta}$.

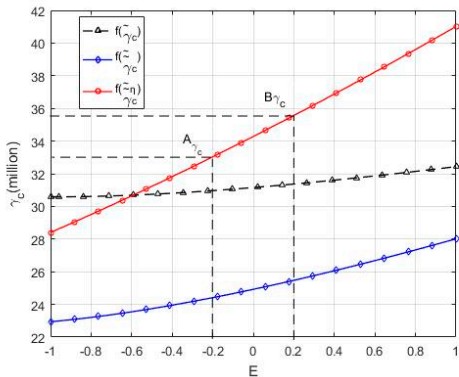

**Figure 6.** Comparison of $\widetilde{\gamma}_c$, $\widetilde{\widetilde{\gamma}}_c$ and $\widetilde{\widetilde{\gamma}}_c^{\eta}$.

The data in Table 8 shows that when $E = 0.2$, the effort level of the private partner decreases by 19%, the project revenue decreases by RMB 5.46 million, which is a decrease of 11.62%; also, its RSR decreased by 6.29%, and the revenue sharing decreased by RMB 6.30 million with a decrease of 19.84%. The comparison of Method #2 with Method #1 shows that, when the project revenue and revenue sharing are both reduced with the reduction of the participants' efforts, the revenue sharing will inevitably drop significantly; and vice versa. Therefore, the link between effort level and revenue sharing can more effectively motivate participants to improve their effort levels for achieving or exceeding the expected revenue of the project.

**Table 8.** Comparison of the revenue sharing of the private partner between Method #1 and Method #2 ($E = 0.2$).

| Indicators | Effort Level | Project Revenue | RSR | Revenue Sharing |
|---|---|---|---|---|
| Method #1 | 1 | 46.98 | 67.64% | 31.78 |
| Method #2 | 0.81 | 41.52 | 61.35% | 25.47 |
| Comparison | −0.19 | −5.46 | −6.29% | −6.30 |
| Comparison (%) | 19% | −11.62% | — | −19.84% |

*5.3. Comparison 3*

As shown in Figure 2, the project revenue remains unchanged. Figures 3 and 4 show that the RSR and the revenue sharing of the government decrease due to the decreased input ratio. As shown in Figures 5 and 6, the RSR and revenue sharing of the private partner increased as a result of the increased input ratio. Compared with Method #2, Method #3 shows that the RSR and revenue sharing are positively correlated with the input ratio.

Table 9 shows that when $E = 0.2$, the input ratio of the government decreases by 0.29, the RSR decreases by 24.31%, and the revenue sharing decreases by RMB 10.09 million, which is a decrease of 62.89%. As shown in Table 4, when $E = 0.2$, the input ratio of the private partner increased by 0.29, the RSR increased by 24.31%, and the revenue sharing increased by RMB 10.09 million, which is an increase of 39.62%. Therefore, the higher the input ratio, the higher the RSR and the more revenue sharing, which can encourage private partners to actively invest in operational TOT projects.

**Table 9.** Comparison of the revenue sharing of the participants between Method #3 and Method #2 ($E = 0.2$).

| Indicators | Government | | | Private Partner | | |
|---|---|---|---|---|---|---|
| | Input Ratio | RSR | Revenue Sharing | Input Ratio | RSR | Revenue Sharing |
| Method #2 | 0.5 | 38.65% | 16.05 | 0.5 | 61.35% | 25.47 |
| Method #3 | 0.21 | 14.34% | 5.96 | 0.79 | 85.66% | 35.56 |
| Comparison | −0.29 | −24.31% | −10.09 | 0.29 | 24.31% | 10.09 |
| Comparison (%) | −58% | - | −62.89% | 58% | - | 39.62% |

### 5.4. Potential Application from the Functional Analysis of Method #3

The fuzzy payoff function of the operational TOT project and fuzzy value of revenue-sharing both have a corresponding fuzzy structure element value. The fuzzy element structure value enables the exact correspondence between the fuzzy payoff function and the revenue-sharing value in different stages of the contract. This approach can be applied to the following two scenarios.

(1) Application in the Contract Design Stage.

If the membership degree of the payoff function is known, the corresponding expected project revenue interval and the revenue-sharing intervals of participants and the RSR interval can be predicted. Therefore, the RSR that is based on the input ratio and the double-fuzzy Shapley value can predict the influence of external uncertainty, effort level uncertainty and the input ratio on the revenue sharing of participants. In other words, it meets the revenue-sharing forecast in the contract design stage. For example, when the membership degree is 0.8, there is $-0.2 \leq E \leq 0.2\ E$, $\widetilde{v}^{\eta}_{\widetilde{TOT}}$ is RMB 37.88–41.52 million, $\widetilde{\widetilde{\beta}}^{\eta}_{g}$ is 12.79–14.34%, $\widetilde{\widetilde{\gamma}}^{\eta}_{g}$ is RMB 4.84–5.96 million, and $\widetilde{\widetilde{\beta}}^{\eta}_{c}$ is 87.21–85.66%, and $\widetilde{\widetilde{\gamma}}^{\eta}_{c}$ is RMB 33.04–35.57 million. As the vertical axis interval corresponding to point A and point B are shown in Figure 2, Figure 3, Figure 4, Figure 5, Figure 6 respectively.

(2) Application in the Contract Execution Stage.

If the operational TOT project revenue is known, the value of the participant's RSR and level of revenue sharing can be calculated according to the corresponding fuzzy element value. That is to say, the RSR based on the input ratio and the double-fuzzy Shapley value can be used for accurate revenue sharing in the contract execution stage. Such as, if $\widetilde{v}^{\eta}_{\widetilde{TOT}}$ is RMB 41.52 million, $E = 0.2$, so the corresponding $\widetilde{\widetilde{\beta}}^{\eta}_{g}$ is 14.34%, $\widetilde{\widetilde{\gamma}}^{\eta}_{g}$ is RMB 5.96 million, $\widetilde{\widetilde{\beta}}^{\eta}_{c}$ is 85.66%, and $\widetilde{\widetilde{\gamma}}^{\eta}_{c}$ is RMB 35.57 million. They are points $B_{\beta g}$, $B_{\gamma g}$, $B_{\beta c}$, $B_{\gamma c}$, as shown in Figures 3–6 respectively.

Therefore, the RSM can be applied to both the revenue-sharing forecast in the contract design phase and the accurate revenue sharing during the contract execution stage. In addition, the above application is an example of the total net profit of the project operational period. Similarly, this RSM can also be applied to the distribution of annual net profit of the operational TOT project in the contract execution period.

### 5.5. Comparison of Different Modified Shapley Value Methods

According to Section 2.3., there are three aspects associated with modifying of the Shapley value, which are fuzzy payoffs, fuzzy alliance, and input ratio coefficient. In addition to the "fairness" of the RSMs based on the modified Shapley value, the following features and applications highlight the improved performance of the method. (1) Flexibility, where revenue sharing changes with income uncertainty, and thus, the revenue-sharing approach is flexible. (2) Incentive, where RSR is positively correlated with effort level and input ratio. (3) Forecasting, where the revenue-sharing interval is forecast according to the change of revenue. (4) Exact distribution, where the participant's revenue sharing is an exact value rather than an interval value.

The methods used by Hu et al. [42], Li et al. [43], Yu et al. [44], Zhang [45] in Section 2.2 are noted as Method 4#, Method 5#, Method 6#, Method 7#. √ indicates the relevant characteristics of an RSM.

The comparison of modifying forms, features and applications in Methods 3#, 4#, 5#, 6# and7# in terms of modifying forms and features are shown in Table 10.

**Table 10.** Comparison of Methods 3#, 4#, 5#, 6# and 7# in terms of modifying forms and features.

| Factors | Method 3# | Method 4# | Method 5# | Method 6# | Method 7# |
|---|---|---|---|---|---|
| Modifying forms | | | | | |
| Fuzzy payoff | √ | | | √ | √ |
| Fuzzy alliance | √ | | | | √ |
| Input ratio | √ | √ | √ | √ | |
| Effort level | √ | √ | | √ | |
| Features | | | | | |
| Flexibility | √ | √ | √ | √ | √ |
| Incentive | √ | √ | √ | √ | |
| Applications | | | | | |
| Forecasting | √ | | | √ | √ |
| Exact distribution | √ | √ | √ | | |

As shown in Table 10, each method has flexibility. Methods 4# and 5# are incentive as they are modified by the input-ratio coefficient. In addition, because they do not take into account uncertainties, the revenue sharing is exact, but they do not have the function of forecasting of revenue-sharing interval. Methods 6# and 7# are meliorated with fuzzy payoffs, which is predictive but not exact. Method 6# modified with input-ratio coefficient is incentive, while method 7# is lack of incentive without using input ratio coefficient. Method 3# is improved systematically based on double fuzzy and input-ratio coefficient. This method is both predictive and exact for triangular fuzzy element function expression. In addition, Method 3# has a better incentive by linking the revenue sharing of participants with the input ratio and the effort level, which promotes the participants to achieve or exceed the expected revenue and IRR of the project and promotes the enthusiasm of private parties to participate in TOT projects.

## 6. Conclusions

This research study has identified the factors impacting revenue sharing from the perspective of influencing the project revenue and the RSR and categorizes the influencing factors according to three main areas: external uncertainty, effort level and input ratio. According to the three indicators, different methods have to be adopted in order to modify the Shapley value and construct an RSM of the operational TOT project based on input ratio and double-fuzzy Shapely value.

The RSM has four components, which are as follows: (1) The fuzzy payoff Shapley value is used to enable the reasonable distribution of project revenue uncertainty, which is caused by external uncertainty among the participants according to the RSR. (2) The influence of effort level and its uncertainty on project revenue and the RSR is clarified based on the fuzzy alliance Shapley value, that is, the effort level of participants is positively correlated with the project revenue and revenue sharing. Further, the project revenue uncertainty that is caused by the effort level and its uncertainty is distributed reasonably. (3) The RSM simplifies the calculation method of the input ratio correction coefficient and solves the influence of the unequal input ratio of the participants on the revenue sharing in order to encourage the private partner to actively invest in operational TOT projects. (4) Finally, the RSM can meet the revenue-sharing forecast in the contract design stage and revenue sharing in the contract execution stage. Therefore, the RSM has the capacity to support a reasonable level of revenue sharing that is matched with risk sharing, resource input and the effort level of participants. This will also minimize the impact of project revenue uncertainty on individual participants, thereby reducing the likelihood of the need for contract renegotiations and help avoid the potential dispute of determining the MRG and MRC. In addition, the RSM emphasizes the effort level linked with project

revenue and participants' revenue sharing, which further encourages participants to enhance their effort levels to achieve or even exceed the expected project revenue.

Analysis of the results of the case study investigation can be considered in terms of the following four main concluding points. (1) The uncertainty of the project revenue can be shared by the participants according to the required proportion, which distributes the revenue uncertainty among the participants. (2) When the project revenue decreases and the RSR increases with the decrease of effort level, the change of revenue sharing is uncertain. Consequently, when the effort level is higher, the project revenue is higher, the revenue sharing increases, and vice versa. Also, when the project revenue and revenue sharing are both reduced with the reduction of the participants' efforts, the revenue sharing will inevitably drop significantly, and vice versa. This highlights that the effort level is positively correlated with the project revenue and revenue sharing. Thus, the project revenue and the revenue sharing all increase as the effort level is improved. This indicates that the effort level linked with revenue sharing can more effectively motivate participants to improve their respective effort levels to achieve the expected revenue of the project or even exceed the expectations. Therefore, the interests of both parties are ultimately maximized. (3) The resource input ratio is positively related to the RSR and the revenue sharing. The higher the input ratio, the higher the RSR and the revenue sharing, which encourage the private partner to actively invest in operational TOT projects. (4) The RSM can be applied to both the revenue-sharing forecast in the contract design phase and the accurate revenue sharing during the project operation period.

In the contract design stage, the expected revenue during franchise period is the project revenue corresponding to the project company's expected minimal IRR. The RSM distributes the revenue uncertainty among the participants proportionally according to the change of expected revenue and forecasts the revenue-sharing interval of the participants. Each participant decides whether to participate in the project based on the forecast results and respective risk attitude. That is, the RSM can provide a basis for participants' investment decisions. As the forecast results have included both less-than-expected and more-than-expected revenue distribution. If the participant decides to participate in the project, it means that it is an acceptance risk of revenue less than is expected for participants. Therefore, the government does not need to assume the MRG, and there is no need to allocate another plan for the revenue more than expected, which means that the RSM is more concise. In the contract execution stage, the RSM conducts exact revenue sharing according to realized revenue which is a point of the prediction interval. Thus, the revenue sharing meets respective expected minimal rate of return.

This RSM is designed for the operational TOT projects based on the characteristics of the operational TOT projects. Therefore, the RSM cannot be directly applied to quasi-operational TOT projects and other types of PPP projects. In future studies and in order to expand the application scope of the RSM, relevant parameters need to be adjusted according to the characteristics of quasi-operational TOT projects or for other types of PPP projects.

**Author Contributions:** Y.D. and J.F. proposed the original idea, as well Y.D. and J.Z. organized the data collection. Y.D. developed the theoretical part while J.F. and Y.K. developed the empirical model. S.P.P. improved English expression. All the authors provided critical feedback and helped shape the framework, analysis and conclusion. All the authors read and approved the manuscript.

**Funding:** This research was funded by [Hubei Provincial Development and Reform Commission of China], grant number [20162s0013]; [National Natural Science Foundation of China], grant number [71301013]; [National Social Science Fund Post-financing projects], grant number [19FJY016]; [Humanity and Social Science Program Foundation of the Ministry of Education of China], grant number [17YJA790091]; and [List of Key Science and Technology Projects in China's Transportation Industry in 2018-International Science and Technology Cooperation Project], grant number [2018-GH-006].

**Conflicts of Interest:** The authors declare no conflicts of interest.

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
