# Peer review of "Developing a Revenue Sharing Method for an Operational Transfer-Operate-Transfer Project"

_sustainability, doi:10.3390/su11226436_

Round 1

Reviewer 1 Report

This manuscript discusses in detail the revenue sharing model within PPP (TOT thereby) projects. The contribution is particularly interesting and well structured. The Authors illustrated “the fuzzy payment Shapley value method for revenue distribution for an operational TOT project” with a deep analytical description and well-organized results.

The only concern is about the missing argumentation on the eventual implication of RSM for the Internal Rate of Return, usually used to evaluate the attractiveness (or grade of profitability) of a project or investment in PPP initiatives.

Probably, both the conclusions and the introduction may briefly argue the connection between RSM and IRR, to offer a more comprehensive picture of the value-added of RSM to IRR limits.

Author Response

The authors wish to extend the thanks to the editors and referees for their support, constructive comments and suggestions. We are very grateful to have the opportunity to improve our work.

We have revised the manuscript accordingly.

Each individual comment has been addressed fully (please see the attachment).

Reviewer 2 Report

The paper offers a specific contribution on revenue sharing.

The paper is very technical for a journal like Sustainability. However the authors do a good job in explain the method in detail.

I think the literature review would benefit from the inclusion of a literature review table that would help you to bring out your contribution with respect to the existing ones. I suggest considering how new revenue techniques such as dynamic pricing can help to maximize revenues and reducing uncertainty. Here a paper from Abrate et al. (2019) in Tourism Management can help.

Methodologically, the Shapley value is of merit. Also here a summary table with the findings (especially given the empirical application) would be helpful for readers.

Author Response

The authors wish to extend the thanks to the editors and referees for their support, constructive comments and suggestions. We are very grateful to have the opportunity to improve our work.We have revised the manuscript accordingly.

Each individual comment has been addressed fully (please see the attachment).
